# Biopsychosocial factors are associated with impaired sexual function in Mexican patients with rheumatoid arthritis

**Loraine Ledón-Llanes**[1]ⓧ, **Irazú Contreras-Yáñez**[2]ⓧ, **Guillermo Arturo Guaracha-Basáñez**[2]ⓧ, **Salvador Saúl Valverde-Hernández**[2]ⓧ, **Maximiliano Cuevas-Montoya**[2]ⓧ, **Ana Belén Ortiz-Haro**[2]ⓧ, **Virginia Pascual-Ramos**[2]ⓧ *

**1** Department of Biology of Reproduction, Instituto Nacional de Ciencias Médicas y Nutrición Salvador Zubirán, Mexico City, Mexico, **2** Department of Immunology and Rheumatology, Instituto Nacional de Ciencias Médicas y Nutrición Salvador Zubirán, Mexico City, Mexico

ⓧ These authors contributed equally to this work.

* virtichu@gmail.com

## Abstract

### Background

Rheumatoid arthritis (RA) is a chronic disease with worldwide representation that impacts every domain of a patient´s life, extending to sexual and reproductive domains. The study characterized sexual health (SH) and reproductive health (RH) in Mexican RA outpatients and identified factors associated with impaired sexual function (ISF).

### Methods

From September 1, 2020—January 31, 2022, consecutive RA participants had semi-structured interviews focusing on their SH and RH biographies, and self-administered questionnaires were applied to assess patient-reported outcomes, including fatigue with the Functional Assessment of Chronic Illness Therapy-Fatigue scale (FACIT-F). ISF was defined based on published cut-offs of the International Index of Erectile Function (IIEF) in males and the Female Sexual Function Index (FSFI) in females (≥1 sexual intercourse in the last four weeks was required for index scoring). Multivariable logistic regression analysis was used to identify the factors associated with ISF.

### Results

There were 268 participants, and 246 (91.8%) were females. Participants had 13 years of disease duration. Among females, 151 (61.4%) had FSFI applied, and the satisfaction domain was impaired in 111 (73.5%). Among males (N = 22), 17 (77.3%) had IIEF applied, and erectile dysfunction was present in 5 (29.4%). Almost half of the participants denied using a family planning method, were in their 50s, and receiving teratogenic drugs; 89.7% of the participants had children.

ISF was detected in 94 (62.3%) females and 3 (17.6%) males. Male sex (aOR: 0.07, 95% CI: 0.01–0.36, p = 0.001), FACIT-F score (aOR: 0.96, 95%CI: 0.92–1.00, p = 0.03), and

**Data Availability Statement:** All relevant data are within the manuscript and its Supporting Information files. The complete data bases are not

openly available, as they compromise patient identification and are considered sensitive information. The study analyzed data from outpatients from a single center in Mexico City (information regarding the single center can be deduced from co-authors' data). Patients' information includes data related to their perceived sexual and reproductive health and rheumatic disease biography. Because of the above reasons, we consider it unethical to put the data in a public repository or supporting information files. Our complete data are available from the corresponding author upon reasonable request and with our local IRB's approval. Data requests can also be placed with the chair of the Research Ethics Committee, Dr. Sergio C. Hernández Jiménez, via e-mail at: sergio.hernandezj@incmnsz.mx.

**Funding:** The author(s) received no specific funding for this work.

**Competing interests:** The authors have declared that no competing interests exist.

cohabitation with the couple (aOR: 0.32, 95%CI: 0.11–0.96, p = 0.04) were associated with ISF.

## Conclusions

We observed a disproportionate burden of ISF among women with RA compared to male participants. Male sex, lesser fatigue, and cohabitation with the couple were protective against ISF. Regardless of the prevalent use of teratogenic medications, contraceptive use was suboptimal among the participants.

## Introduction

Rheumatoid arthritis (RA) is one of the most common immune-mediated diseases, with a remarkably consistent prevalence worldwide, at about 0.5% to 1% [1]. The disease has a female preponderance. In Caucasians, it is two to three times as common among females as males and can occur at any age, but the incidence peaks in the third through the fifth decades of life [1]. RA is characterized by pain, systemic inflammation, progressive articular damage, and extra-articular manifestations. If uncontrolled, it might cause disability and impact every domain of a patient´s life [2], extending to sexual and reproductive domains [3].

Sexual and reproductive health is a complex construct, integrated by two dimensions, sexual health (SH) and reproductive health (RH), that are recommended to be addressed separately [4]. Moreover, the World Health Organization (WHO) recognized SH as an umbrella term that includes RH [5]. SH and RH are greatly influenced by the social, cultural, educational, and economic conditions of the community where people are born and live [4]. Systematic literature reviews have consistently revealed an impact of RA on SH and RH in both women and men [6–9]. Meanwhile, research on attribution to the dysfunction of either dimension among RA patients has shown conflicting results. Impaired SH has been related to RA symptoms and reduced patient function [10–19], the long and chronic course of the disease [20,21], comorbid conditions with emphasis on mental health comorbidities [19,22–25], the medications used [26,27], and hormone imbalance [28,29]. RH in RA patients has focused on reduced fertility [30–35], which has been related to the disease process itself [30,33–35], the therapy [29], gonadal dysfunction [36], physicians' advice to patients [30,31], and patients own decisions [30,31]. A higher frequency of adverse outcomes has also been described in pregnancies that occurred after RA onset, compared to the pre-RA obstetric path [37].

RA presents particular characteristics in the Latin American region, which are not limited to a younger age at presentation and extreme female preponderance compared to Caucasian populations [38]. Nationality and ethnicity also influence patient perceptions and views regarding the RA priority domains, concerns, and interests [39,40], the ways SH and RH are conceptualized [41], and the preferred patient-doctor relationship [42]. These characteristics shape patients' experiences and how they are communicated to physicians.

Finally, the Corona Virus Disease 2019 (COVID-19) pandemic has threatened people´s physical and mental health worldwide, including patients with rheumatic diseases [43]. Additionally, the pandemic has led to issues in public health ethics. The need to serve patients with COVID-19 has translated into rationing the care of patients with chronic conditions, which has aggravated existing disparities [44]. World regions where public life was characterized by fragile health systems and long-standing and pervasive inequity, such as Latin America, have faced a humanitarian crisis [45]. Besides, the lockdown produced deep and abrupt changes in

personal, family, and social life, increasing distress and intra-family violence, particularly in women and in Mexico compared to other countries [46]. Because of the close relationship between psychological components and sexual function, several studies have shown decreased sexual activity during the social restriction period due to the COVID-19 pandemic [47].

Considering the above arguments, the study aimed to comprehensively characterize SH and RH in Mexican RA outpatients and identify relevant factors associated with impaired sexual function (ISF). We were particularly interested in determining disease activity and COVID-19 pandemic-related characteristics.

## Materials and methods

### Setting and participants

The INCMyN-SZ is a quaternary care national referral center for rheumatic diseases in Mexico City. Participants were identified from the Department of Immunology and Rheumatology outpatient clinic. Consecutive RA patients waiting for a scheduled consultation were invited to participate. RA diagnosis was based on the treating rheumatologist's criteria (all were board-certified). Exclusion criteria were RA patients with overlapping rheumatologic syndrome (except Secondary Sjögren Syndrome) and patients with uncontrolled comorbidity requiring treatment intensification or palliative care.

### Study design, study assessments, and data collection

This cross-sectional study was conducted between September 1, 2020, and January 31, 2022. STROBE´s guidelines were followed (Please refer to the SI "STROBE checklist for cross-sectional studies," S1 Appendix).

All the participants had a 30–60 minutes-duration semi-structured interview using an interview guide to assess the following aspects: socio-demographics (including partner-related information), relevant information related to current and past psychiatric and medical conditions, current treatments, legal/illegal substance use (including psychopharmaceuticals) and an extensive SH and RH biographies which included the exploration of the participants' perception about the effects of the COVID-19 pandemic on their lives.

After that, self-administered Spanish versions of the following questionnaires (paper format) were provided to the participants: the Hospital Anxiety and Depression Scale (HADS) [48], the Routine Assessment of Patient Index Data 3 (RAPID-3) to assess disease activity/severity [49], the Euro Qol-5 dimensions (EQ-5D) to evaluate health related-quality of-Life [50], the Functional Assessment of Chronic Illness Therapy-Fatigue (FACIT-F) [51], the International Index of Erectile Function (IIEF) in male participants and the Female Sexual Function Index (FSFI) in female participants [52]. Also, considering the study's inclusion period, a self-administered COVID-19 survey to address healthcare interruption, individuals´ perception of the pandemic seriousness in the country, participants' risk perception of SARS-CoV-2 infection, participants follow-up of physical distancing recommendation, the family economic and relational impact attributed to COVID-19 pandemic and ten negative emotions attributed to the COVID-19 pandemic was filled out by the participants [53] (Please refer to the S2 "COVID-19 Survey", S2 Appendix). Interviews and questionnaires administration were scheduled in a private room.

Finally, participants had their charts reviewed by a data abstractor who used standardized formats to confirm sociodemographic information, RA-related characteristics, treatment, and Charlson score [54]. Only the person responsible for the data collection had access to information that could identify individual participants; none of this information was transferred to the database, so no research team member had access to it after this process.

## Measures

**ISF in male participants.**   It was diagnosed based on the IIEF score. The IIEF is a multi-dimensional self-report instrument for evaluating male sexual function over the previous four weeks, which focuses on erectile function/dysfunction (ED). The IIEF includes 15 items distributed into five relevant domains of male sexual function: erectile function (five items), orgasmic function (two items), sexual desire (two items), intercourse satisfaction (three items), and overall satisfaction (two items). For each sexual domain, a score is calculated, and the six domain scores can be added to obtain a score from 5–75, with higher scores indicating better sexual function. The total "erectile function" score ranges from 6–30 with a cut-off for ED<26. Additional cut-off values were used for the other sexual domains based on a qualitative analysis in which the diminished and the presence of difficulties were interpreted as impairing indicators [55]. Therefore, each domain under 60% score (<6 in orgasmic function, sexual desire, overall satisfaction, each, and <9 in intercourse satisfaction) was considered impaired. The cut-off value for the total sexual function score was determined by adding the cut-off values of each sexual function domain. Hence, a total IIEF score of <53 was considered ISF. At least one sexual intercourse in the last four weeks was required to provide an IIEF score.

**ISF in female participants.**   It was diagnosed based on the FSFI score. The FSFI is a 19-item self-report inventory designed to assess critical domains of female sexual function over the previous four weeks. Items are distributed into six domains: desire (two items), arousal (four items), lubrication (four items), orgasm, satisfaction, and pain (three items each). The FSFI total score is the sum of the six domain scores and ranges from 2 to 36. Higher scores indicate better functioning. In clinical practice, an FSFI cut-off score of <26.55 has been widely used to define ISF, while a score <3.9 is considered an impairment for each sexual function domain [21,56]. At least one sexual intercourse in the last four weeks was mandatory to score the IIEF.

**RA-disease activity categories.**   They were defined based on RAPID-3 scores [49]. The RAPID-3 includes three measures: physical function, pain, and a patient global estimate evaluation. It has a raw score of 0–30 and an adjusted score of 0–10, with higher scores translating into higher disease activity/severity. Four proposed categories are defined based on 0–30 scale cut-offs: >12 as high disease activity, 6.1–12.0 as moderate disease activity, 6.0–3.1 as low disease activity, and ≤three near-remission.

*Sexual discomforts*

Sexual expressions that, although they do not constitute sexual dysfunction according to the clinical criteria (APA 2013), have negative effects on sexual well-being from the patient´s subjective assessment [57].

## Sample size calculation

Previous studies have found disease activity parameters (pain, fatigue, erythrocyte sedimentation rate, and disease activity score) and impaired function as significant determinants for ISF [10,12,14,17,22]. We selected the RAPID-3, which accounts for disease activity and severity. We arbitrarily hypothesized an effect size of odds ratio (OR) 2.5 in a no-normal distribution. We estimated the sample size using a one-tailed test, 5% significance level, and 80% power. We obtained (at least) 151 patients. The final sample size and the participants´ distribution gave us 80% power for a one-tailed test to detect an effect size of 0.29.

## Statistical analyses

We performed a descriptive statistical analysis, presenting frequencies for categorical variables and median (interquartile range, [IQR]) for numerical variables.

Characteristics of participants with ISF were compared to those of their counterparts using appropriate tests. The Mann-Whitney U test was used to compare continuous variables between two groups when they did not show a normal distribution (Kolmogorov-Smirnov). Fisher's exact test or $X^2$ test was used to compare proportions.

Multivariable logistic regression analysis was performed to identify factors associated with ISF. We initially conceived a global model where variables' inclusion was based on their statistical significance in the univariable analysis (p≤0.10) and/or their clinical relevance; in particular, RAPID-3 was forced into the models tested. A test-based backward selection was used to define the final model. Correlations between variables were examined to avoid overfitting the models, and when relevant (rho>0.70), only one variable was selected. The Nagelkerke pseudo-$R^2$ test is reported as a measure of model fit goodness. Results are expressed as OR and adjusted OR (aOR) (exponentiated regression coefficients, exp[$\beta$] and their 95% confidence interval). The following variables were considered for aOR: age, sex, being in a relationship, time in a relationship, cohabitation with the couple, FACIT-F score, the self-care and usual activities domains of the EQ-5D, past sexual discomforts, RAPID-3 score, and infertility diagnosis. A value of p≤0.05 was considered statistically significant.

Missing data varied from 0 to 20% of the individual items, and no imputation was performed. Reasons for missing data included participants skipping answers and patients selecting the "I do not want to answer" option.

All statistical analyses were performed using Statistical Package for the Social Sciences version 21.0 (SPSS Chicago IL).

### Ethics

The Research Ethics Committee of the Instituto Nacional de Ciencias Médicas y Nutrición Salvador Zubirán (INCMyN-SZ) approved the study (reference number: IRE-3388-20-21-1). All the patients included provided written informed consent.

## Results

### Overall population characteristics

During the study period, we included 268 RA participants. **Table 1** summarizes participants' characteristics, including socio-demographics, RA-related and mental-health-related features, and COVID-19-related information. The distribution among female and male participants is also presented.

### SH in the entire population

There were 11 out of 253 (4.3%) participants who reported they had past sexual discomforts (15 data unavailable [U]), while 52 out of 179 (29.1%) reported current sexual discomforts (89 U), among whom only 5 (2.8%) received related interventions.

The majority of the participants (245 [91.4%]) had the sexual desire domain scored, and 168 (62.7%) referred to at least one sexual intercourse in the previous four weeks, which was required to score indices related to sexual function.

Among female participants (N = 246), 151 (61.4%) have the FSFI scored, and the median global score was under the cut-off, 25 (22–28). **Fig 1** summarizes the distribution of impaired FSFI domains. ISF was present in 94 (62.3%) female participants.

Among male participants (N = 22), 17 (77.3%) scored the IIEF, and the median global score was 64 (57–70). **Fig 1** presents the distribution of impaired IIEF domains. ISF was present in only 3 participants (17.6%).

**Table 1. Socio-demographics, RA-related, mental-health-related, and COVID-19-related characteristics in the participants enrolled in the study, female and male participants.**

| Variable<br>*category* (n1/n2/n3) | All participants | Female participants | Male participants |
|---|---|---|---|
| **Socio-demographic characteristics** | | | |
| Females (268/246/22) | 246 (91.8) | 246 (100) | NA |
| Years of age* (268/246/22) | 47 (38–53) | 47 (38–53) | 42 (34.8–52.3) |
| Years of Scholarship* (268/246/22) | 12 (9–16) | 12 (9–16) | 12 (9–16) |
| Occupation<br>*Formal job (268/246/22)*<br>*Non-formal job (268/246/22)*<br>*Household work (268/246/22)* | 98 (36.1)<br>44 (16.4)<br>119 (44.4) | 91 (37)<br>32 (13)<br>119 (48.4) | 7 (31.8)<br>12 (54.5)<br>0 |
| Religious beliefs<br>*Catholics (268/246/22)*<br>*Non-religious (268/246/22)* | 198 (73.9)<br>38 (14.2) | 180 (73.2)<br>34 (13.8) | 18 (81.8)<br>4 (18.2) |
| Relationship<br>*Being in a relationship (268/246/22)*<br>*Years in a relationship[1]* (268/246/22)*<br>*Stable bond[1] (268/246/22)*<br>*Cohabitation with the couple[1] (268/246/22)* | 191 (71.3)<br>18 (6–30)<br>170 (89)<br>148 (77.5) | 172 (69.9)<br>15.8 (6–30)<br>153 (88.9)<br>134 (77.9) | 19 (86.4)<br>18 (6–30)<br>17 (89.5)<br>14 (73.7) |
| **RA-related characteristics** | | | |
| Years of disease duration* (268/246/22) | 13 (8–18) | 13 (8–18) | 10 (4.5–15.5) |
| Charlson score* (268/246/22) | 1 (1–1) | 1 (1–1) | 1 (1–1) |
| Comorbidities (268/246/22) | 143 (53.4) | 132 (53.7) | 11 (50) |
| Use of prednisone (268/246/22) | 95 (35.4) | 87 (35.4) | 8 (36.4) |
| N° DMARDs/patient* (268/246/22) | 1 (1–2) | 1 (1–2) | 1 (1–2) |
| RAPID 3 score (0–30)* (268/246/22) | 7.7 (2.3–14.3) | 7.9 (2.3–14.3) | 3.8 (0.8–10.8) |
| Low disease activity and near remission (RAPID3 score ≤6) (268/246/22) | 115 (42.9) | 101 (41.1) | 14 (63.6) |
| FACIT-F score* (0–52) (268/246/22) | 11 (5–20) | 11 (5–20) | 10.5 (3.5–17) |
| EQ-5D (No problems)<br>*Mobility (245/226/19)*<br>*Self-care (245/226/19)*<br>*Usual activities (245/226/19)*<br>*Pain and discomfort (245/226/19)*<br>*Anxiety and depression (245/226/19)* | 143 (58.4)<br>181 (73.9)<br>139 (56.7)<br>74 (30.2)<br>138 (56.3) | 131 (58)<br>166 (73.5)<br>128 (56.6)<br>67 (29.6)<br>127 (56.2) | 12 (63.2)<br>15 (78.9)<br>11 (57.9)<br>7 (36.8)<br>11 (57.9) |
| **Mental health-related characteristics** | | | |
| Mental health comorbidity (256/238/18) | 32 (12.5) | 31 (13) | 1 (5.6) |
| Legal/illegal substances use/abuse (268/246/22) | 24 (9) | 17 (6.9) | 7 (31.8) |
| Psychotropic drugs (230/214/16) | 16 (7) | 16 (7.5) | 0 |
| HADS score*<br>*Depression score (0–21) (268/246/22)*<br>*Anxiety score (0–21) (268/246/22)* | 3 (1–6)<br>5 (3–8) | 3 (1–6)<br>5 (3–8) | 2.5 (1–4.3)<br>5 (3–8) |
| Abnormal HADS score (>11) (268/246/22) | 99 (36.9) | 93 (37.8) | 6 (27.3) |
| **COVID-19-related characteristics** | | | |
| Healthcare interruption (257/237/20) | 174 (67.7) | 158 (66.7) | 16 (80) |
| RA-related treatment changes because of the COVID-19 pandemic (252/232/20) | 132 (52.4) | 125 (53.9) | 7 (35) |
| Patients´ follow-up of recommendations to contain virus spread (Always/Most of the time) (253/234/19) | 207 (81.8) | 192 (82.1) | 15 (78.9) |
| Patient's perception of the pandemic seriousness in Mexico (Very high/ High) (252/234/18) | 232 (92.1) | 215 (91.9) | 17 (94.4) |
| Patient´s risk perception of Sars-Cov-2 infection (High/Very High) (216/202/14) | 92 (42.6) | 86 (42.6) | 6 (42.9) |
| Feeling anxious (251/233/18) | 79 (31.5) | 74 (31.8) | 5 (27.8) |
| Feeling worried (253/235/18) | 115 (45.5) | 108 (46) | 7 (38.9) |
| Feeling fearful (250/232/18) | 85 (34) | 79 (34.1) | 6 (33.3) |
| Feeling alertness (257/235/18) | 144 (56) | 135 (57.4) | 9 (50) |

*(Continued)*

**Table 1.** (Continued)

| Variable<br>*category* (n1/n2/n3) | All participants | Female participants | Male participants |
|---|---|---|---|
| Feeling depressed (251/233/18) | 43 (17.1) | 42 (18) | 1 (5.6) |
| Feeling confused (250/233/17) | 49 (19.6) | 46 (19.7) | 3 (17.6) |
| Feeling alarmed (252/234/18) | 81 (32.1) | 77 (32.9) | 4 (22.2) |
| Feeling isolated (252/234/18) | 102 (40.5) | 97 (41.5) | 5 (27.8) |
| Feeling discriminated against (250/232/4) | 13 (5.2) | 13 (5.6) | 0 |
| Feeling bored (246/228/18) | 58 (23.6) | 54 (23.7) | 4 (22.2) |
| Negative family economic impact attributed to the COVID-19 pandemic (232/214/18) | 184 (79.3) | 170 (79.4) | 14 (77.8) |
| COVID-19's negative impact on family members' relationships (233/215/18) | 123 (52.8) | 115 (53.5) | 8 (44.4) |
| Intercourse-related activity modifications during the COVID-19 pandemic (233/213/20) | 81 (34.7) | 74 (34.7) | 7 (35) |

Data are presented as n (%) unless *represents the median (IQR). n1 = data available in all the participants; n2 = data available in female participants; n3 = data available in male participants. [1]Among those with the condition. DMARDs = Disease modifying anti-rheumatic drugs. RAPID-3 = Routine Assessment of Patient Index Data 3. FACIT-F = Functional Assessment of Chronic Illness Therapy–Fatigue. EQ-5D = Euro Qol 5 dimensions. HADS = Hospital Anxiety and Depression Scale (HADS). NA = Not applicable.

### RH in the entire population

In the whole population, 204 (79.7%) participants (12 data U) mentioned past use of a family planning method, while only 128 out of 258 (49.6%) mentioned current usage. Among them, the condom was the most frequently used by 44 (34.3%) participants, 43 (33.5%) mentioned tubal ligation, and 16 (12.5%) intrauterine implants. Interestingly, participants who denied using a family planning method had a median (IQR) of 49.5 (41–55) years of age (vs. 43 [36–49] years of age in the participants with current usage of a family planning method), were indicated DMARDs (124 [95.4%]) and most frequently methotrexate (91 [73.4%]), while only 9 (7%) participants had infertility diagnosis.

Also, 182 females out of 246 (74%) reported pregnancies, with a median (IQR) of 2 (2–3) pregnancies. Overall, 183 participants out of 204 (89.7%) had children with 2 (1–3) children/participant.

Finally, past and current sexually transmitted infections (STI) were reported by 41 in 251 participants (16.3%) and 2 in 223 (2.7%), respectively, and papillomavirus infections were the most prevalent.

### Comparison of participants with and without ISF

Overall, 168 participants met the criteria to score the indices. Their characteristics were compared to those of the 100 participants to whom the index did not apply. Overall, participants from both groups were similar, but being in a relationship in which participants were more frequently referred from the former group (147/168 [87.5%] vs. 44/100 [44%], p≤0.001), in addition to having a stable bond (135/146 [92.5%] vs. 35/44 [79.5%], p≤0.001), having current sexual discomforts (41/115 [35.7%] vs. 11/64 [17.2%], p = 0.01) and a High/Very high-risk perception of SARS-Cov-2 infection (65/136 [47.8%] vs. 27/80 [33.8%], p = 0.05).

Overall, 97 (57.7%) participants had ISF, and their sociodemographic, RA-related, mental health-related, and COVID-19 characteristics were compared to their counterparts (N = 71 [42.3%]) and are summarized in **Table 2** (the p-value is derived from the comparison of the characteristics between participants with ISF and their counterparts, using appropriate tests).

Participants from the former group also mentioned less frequent past sexual discomforts (30/91 [33%] vs. 34/67 [50.7%], p = 0.03), while current sexual discomforts (22/67 [33.8%] vs.

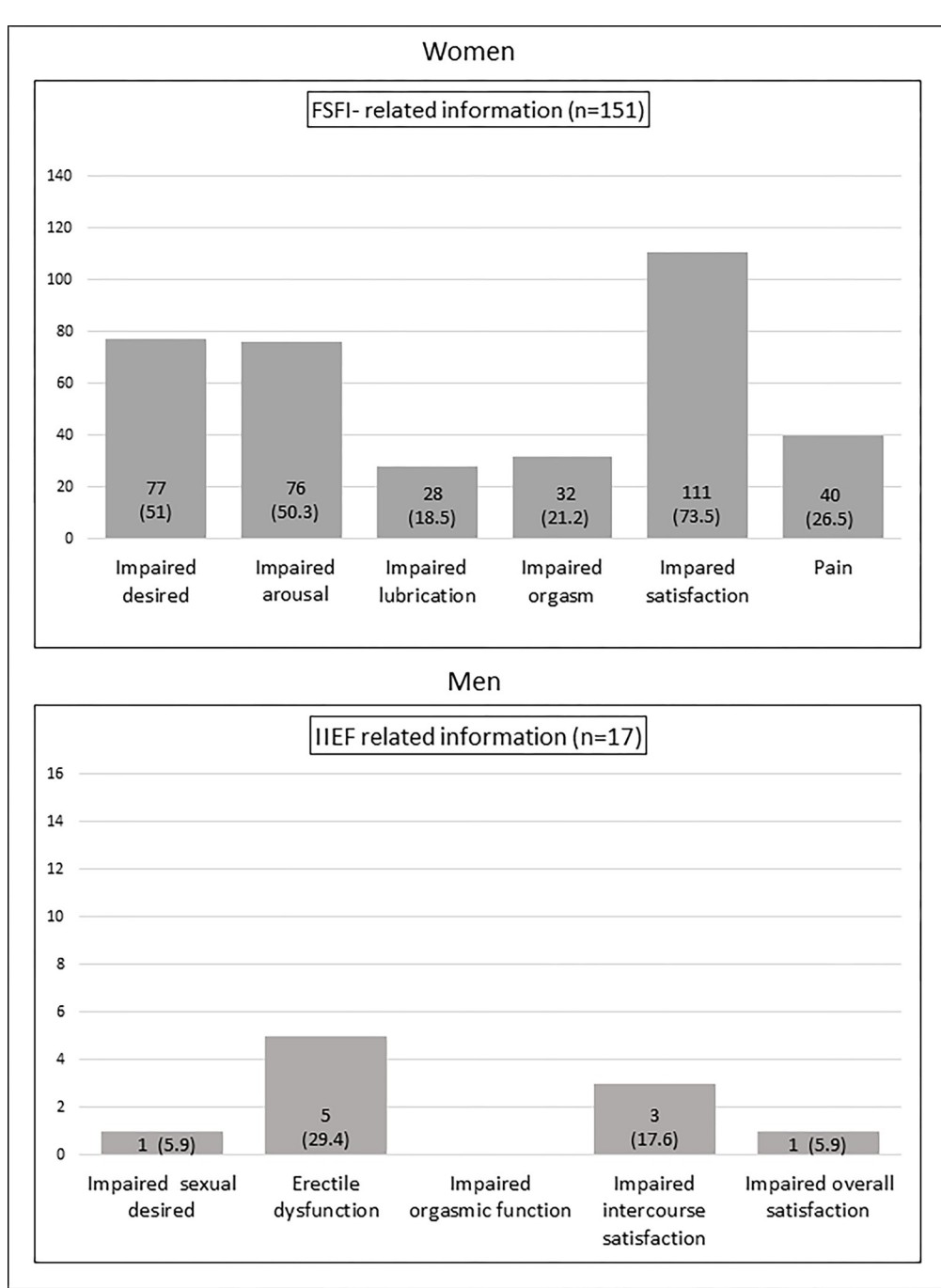

**Fig 1. Sexual function indices-related information in females and males with RA.** Data are presented as the N° of participants (%).

19/49 [38.8%], p = 0.56) and related treatments (2/27 [7.4%] vs. 0, p = 0.49) were similar. As expected, (median, IQR) SF-index scores differed between groups (22.7 [18.4–24.5] in females with ISF vs. 30 [27.8–31.5] in counterpart females and 44 [39–51] in males with ISF vs. 66 [62–71] in counterpart males, p≤0.001 for both). Also, in male and female subpopulations, there was a higher proportion of participants with impaired IIEF and FSFI-specific domains among those with ISF (**Fig 2**).

**Table 2. Comparison of sociodemographic, RA-related, mental health-related, and COVID-19-related characteristics between participants with ISF and their counterparts.**

| Variable<br>*category* (n1/n2) | Participants with ISF | Participants without ISF | p |
|---|---|---|---|
| **Socio-demographic characteristics** | | | |
| Females (97/71) | 94 (96.9) | 57 (80.3) | **0.001** |
| Years of age* (97/71) | 47 (36–52.5) | 46 (38–53) | 0.67 |
| Years of Scholarship* (97/71) | 12 (9–16) | 12 (9–16) | 0.15 |
| Formal/Non-formal job (97/71) | 49 (50.5) | 35 (49.3) | 1 |
| Religious beliefs (97/71) | 79 (81.4) | 63 (88.7) | 0.28 |
| Relationship<br>*Being in a relationship* (97/71)<br>*Years in a relationship[1]** (97/71)<br>*Stable bond[1]* (97/71)<br>*Cohabitation with the couple[1]* (97/71) | 81 (83.5)<br>14 (3–28)<br>75 (92.6)<br>57 (70.4) | 66 (93)<br>20 (9–39)<br>60 (84.5)<br>56 (84.8) | 0.10<br>0.10<br>1<br>**0.02** |
| **RA-related characteristics** | | | |
| Years of disease duration* (97/71) | 12 (8–16) | 14 (18–20) | 0.32 |
| Charlson score* (97/71) | 1 (1–1) | 1 (1–1) | 0.33 |
| Comorbidities (97/71) | 50 (51.5) | 34 (47.9) | 0.26 |
| Use of prednisone (97/71) | 39 (40.2) | 22 (31) | 0.26 |
| N° DMARDs/patient* (97/71) | 1 (1–2) | 1 (1–2) | 0.35 |
| RAPID 3 score (0–30)* (97/71) | 6.5 (1.7–14.3) | 8 (3.6–15) | 0.39 |
| Low disease activity and near remission (RAPID3 score ≤6) (97/71) | 47 (48.5) | 29 (40.8) | 0.35 |
| FACIT-F score* (0–52) (97/71) | 10 (4.5–18) | 13 (8–20) | **0.05** |
| EQ-5D (No problems)<br>*Mobility (95/66)*<br>*Self-care (95/66)*<br>*Usual activities (95/66)*<br>*Pain and discomfort (95/66)*<br>*Anxiety and depression (95/66)* | 61(64.2)<br>78 (82.1)<br>64 (67.4)<br>34 (35.8)<br>58 (61.1) | 38 (57.6)<br>44 (66.7)<br>34 (51.5)<br>156 (22.7)<br>37 (56.1) | 0.42<br>**0.04**<br>**0.04**<br>0.19<br>0.77 |
| **Mental health-related characteristics** | | | |
| Mental health comorbidity (96/68) | 15 (15.6) | 9 (13.2) | 0.82 |
| Legal/illegal substances use/abuse (97/71) | 8 (8.2) | 8 (11.3) | 0.47 |
| Psychotropic drugs (84/64) | 8 (9.5) | 3 (4.7) | 0.27 |
| HADS score*<br>*Depression score (0–21) (97/71)*<br>*Anxiety score (0–21) (97/71)* | 3(1–5)<br>6 (3–8) | 4 (1.8–6.3)<br>5 (3–8) | 0.12<br>0.84 |
| Abnormal HADS score (>11) (97/71) | 34 (35.1) | 28 (39.4) | 0.63 |
| **COVID-19-related characteristics** | | | |
| Health care interruption (93/69) | 58 (62.4) | 47 (68.1) | 0.55 |
| RA-related treatment changes because of the COVID-19 pandemic (89/68) | 43 (48.3) | 38 (55.9) | 0.42 |
| Patients´ follow-up of recommendations to contain virus spread (Always/Most of the time) (90/68) | 75 (81.5) | 58 (85.3) | 0.67 |
| Patient's perception of the pandemic seriousness in Mexico (Very high/ High) (77/59) | 84 (93.3) | 59 (86.8) | 0.18 |
| Patient´s risk perception of Sars-Cov-2 infection (High/Very High) (77/59) | 36 (46.8) | 29 (49.2) | 0.87 |
| Feeling anxious (92/65) | 27 (29.3) | 23 (35.4) | 0.49 |
| Feeling worried (92/67) | 41 (44.6) | 34 (50.7) | 0.52 |
| Feeling fearful (92/65) | 33 (35.9) | 22 (33.8) | 0.87 |
| Feeling alertness (92/67) | 59 (64.1) | 40 (59.7) | 0.74 |
| Feeling depressed (92/66) | 17 (18.5) | 10 (15.2) | 0.67 |
| Feeling confused (92/66) | 17 (18.5) | 13 (19.7) | 0.84 |
| Feeling alarmed (92/67) | 29 (31.5) | 19 (28.4) | 0.73 |
| Feeling isolated (92/66) | 43 (46.7) | 25 (37.9) | 0.33 |

*(Continued)*

**Table 2.** (Continued)

| Variable<br>*category* (n1/n2) | Participants with ISF | Participants without ISF | p |
|---|---|---|---|
| Feeling discriminated against (92/66) | 6 (6.5) | 3 (4.5) | 0.74 |
| Feeling bored (92/64) | 24 (26.1) | 17 (26.6) | 1 |
| Negative family economic impact attributed to the COVID-19 pandemic (88/61) | 68 (77.3) | 47 (77) | 1 |
| COVID-19's negative impact on family members' relationships (89/61) | 40 (44.9) | 25 (41) | 0.29 |
| Intercourse-related activity modifications during the COVID-19 pandemic (68/57) | 31 (45.6) | 23 (40.4) | 0.48 |

Data are presented as n (%) unless *represents the median (IQR). n1 = data available in participants with ISF; n2 = data available in participants without ISF. [1]Among those with the condition. DMARDs = Disease modifying anti-rheumatic drugs. RAPID-3 = Routine Assessment of Patient Index Data 3. FACIT-F = Functional Assessment of Chronic Illness Therapy–Fatigue. EQ-5D = Euro Qol 5 dimensions. HADS = Hospital Anxiety and Depression Scale (HADS).

Finally, five RH characteristics were considered relevant to sexual function. They were compared between both groups: menstruation in the previous year (restricted to females), current use of a contraception method, perception of fertility hinders, infertility diagnosis, and a current reproductive project. Participants with ISF were similar to their counterparts, but the former participants had more frequent infertility diagnoses than their counterparts: 8 out of 71 participants (11.3%) vs. 3 out of 97 (3.1%), p = 0.05.

## Factors associated with ISF

Different models were tested and yielded similar results. **Table 3** presents OR and aOR, 95% CI, and p values for all the variables included in the model. **Fig 3** summarizes the variables that ended up being significantly associated with ISF: the FACIT-F score (protective), cohabitation with the couple (protective), and male sex (protective) ($R^2$ = 0.22).

## Discussion

In the current study, we first observed that the vast majority of female participants had ISF, up to 61.4%, while it was present in only 17.6% of male participants. In females, sexual function domains more frequently affected were satisfaction, desire, and arousal, while erectile function and intercourse satisfaction were more frequently impaired in males. Second, almost half of the participants of the study affirmed using a family planning method, while non-users were in their 50s and primarily receiving teratogenic drugs. Also, most of the participants had more than one child. STIs, primarily papillomavirus, were present in a minority of the participants. Third, we identified three factors that were protective against ISF, namely, male sex, FACIT-F score, and cohabitation with the couple.

Our prevalence of ISF among women with RA and the pattern of the domains more frequently affected is consistent with that reported in reviews and original articles [6,14,58–60]. In a study of 100 sexually active Mexican women with RA and 100 healthy controls, the authors found that up to 49% of the patients experienced ISF based on the FSFI score. Similar to our findings, 50% of the patient participants experienced arousal impairment, although the overall pattern of sexual dysfunction differed from that observed in our patients. These differences could be due to the fact that our patients were older, more frequently active workers, and had their disease under control compared to those described by Rojo-Contreras et al. [60]. These differential characteristics have been shown to impact sexual function. However, our results on male participants differed from those published in a systematic review, where ISF was observed in 33% to 62% of the participants, and a more significant impact on sexual function in males was highlighted [7]. Physical function and disease activity play a more conclusive

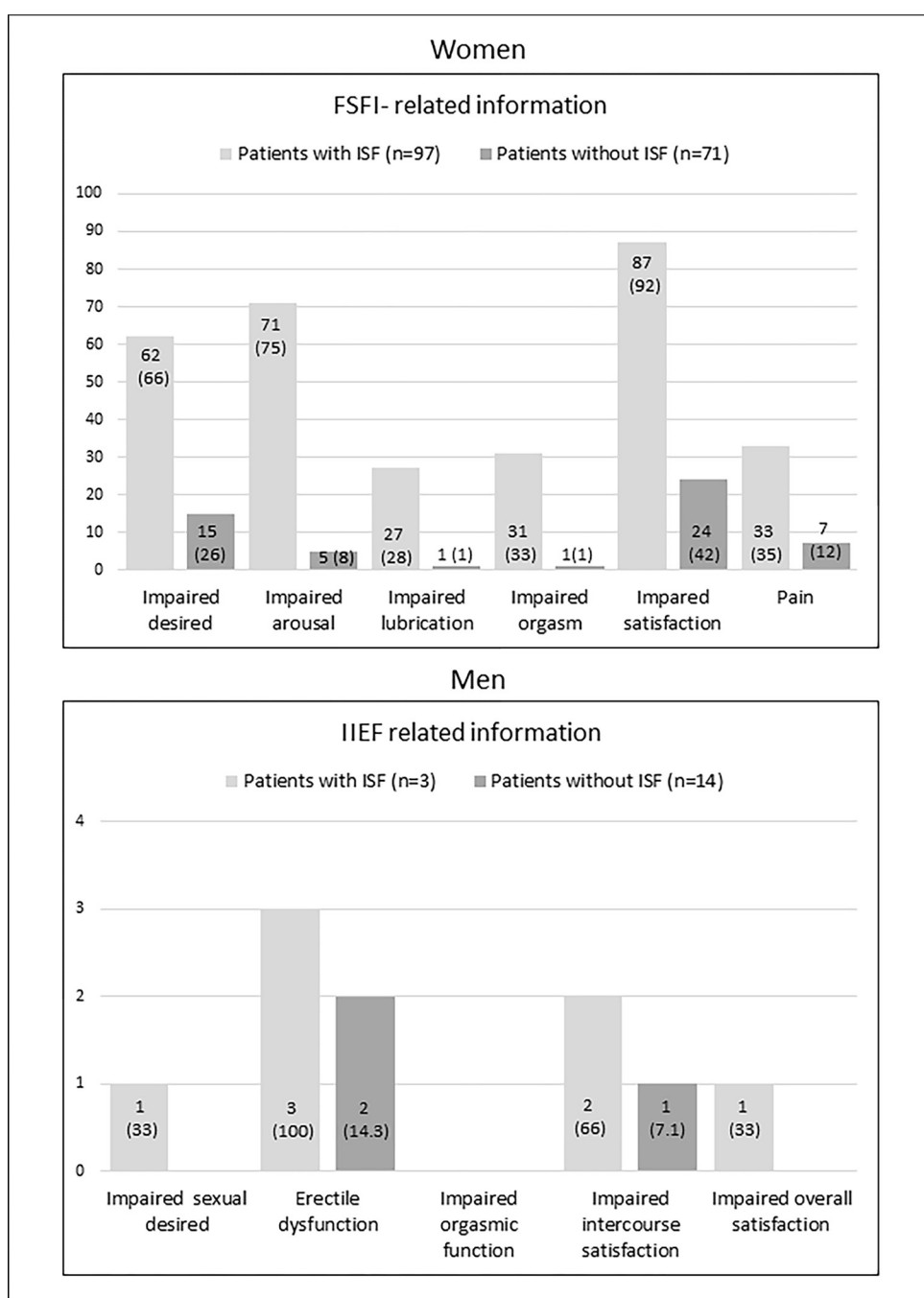

**Fig 2. Comparison of ISF indices-related domains between patients with ISF and their counterparts.** Data presented as N° of patients (%). ISF = Impaired Sexual Function.

role among men than women in sexual function [13,61]. Our male participants scored low on the physical function score of the RAPID3 (median 0.5 [0–3.1]), which translates into a better function and might impede visualizing the intrusiveness of the disease itself in our males´ sexual function. Also, male participants showed few impaired domains, which might have contributed to a better global sexual function score [62].

**Table 3. OR and aOR (95% CI and p-value) for all the variables included in the multivariable logistic regression analysis.**

|  | OR, (95% CI), p | aOR, (95% CI), p |
|---|---|---|
| Age (years) | 1.00, (0.97–1.02), 0.74 | 1.01, (0.92–1.12), 0.86 |
| Male sex | 0.13, (0.04–0.47), 0.002 | 0.20, (0.02–1.70), 0.14 |
| Being in a relationship | 0.38, (0.14–1.10), 0.08 | 0.35 (0.21–1.51), 0.11 |
| Time in a relationship (years) | 0.98, (0.96–1.01), 0.17 | 1.01, (0.93–1.08), 0.93 |
| Cohabitation with the couple | 0.04, (0.13–0.86), 0.02 | 0.51, (0.07–3.92), 0.51 |
| FACIT-F score | 0.98, (0.95–1.01), 0.10 | 0.91, (0.83–1.01), 0.08 |
| Self-care domain of EQ-5 | 0.44, (0.21–0.91), 0.03 | 0.34, (0.06–1.95), 0.23 |
| Usual activities domain of EQ-5 | 0.52, (0.27–0.98), 0.044 | 0.85, (0.14–4.86), 0.86 |
| Past sexual discomforts | 0,48, (0.25–0.91), 0.03 | 0.79, (0.23–2.72), 0.71 |
| RAPID-3 score | 0.99 (0.95–1.03), 0.52 | 1.14, (0.98–1.34), 0.10 |
| Infertility diagnosis | 0.43, (0.14–1.37), 0.15 | 0.28, (0.04–1.79), 0.18 |

Previous reports have confirmed a high prevalence of lack of contraception among child-bearing ability patients with rheumatic diseases [63,64], which has been explained by rheumatologists rarely addressing pregnancy issues [65–67] and the multiple known barriers to contraceptive access [68]. In addition, an increased rate of infertility is currently assumed in RA females and males. We were intrigued by the family size of our participants, which does not substantially differ from that observed in Mexicans [69] and might be related to age at disease diagnosis, around their forties, when decisions to have children have already been made. The participants in our study reported a 16.3% prevalence of past and current sexually transmitted infections (STIs), mostly papillomavirus infections. Previous studies have shown varying prevalence rates among Mexican women with RA [60,70], which could be attributed to differences in the criteria used to diagnose STIs (self-reported vs. physician-established) and the widespread vaccination efforts in recent decades, especially in urban areas [71].

Sex, a biological variable, contributes to several pathogenic and epidemiologic aspects of RA, generating significant differences between affected males and females, which might explain why the male sex was protective against ISF. Overall, RA seems more severe in women, in whom more frequent comorbidities that impact sexual function, such as fibromyalgia and depression, are frequently observed [72]. Moreover, the use of common drugs for treating RA

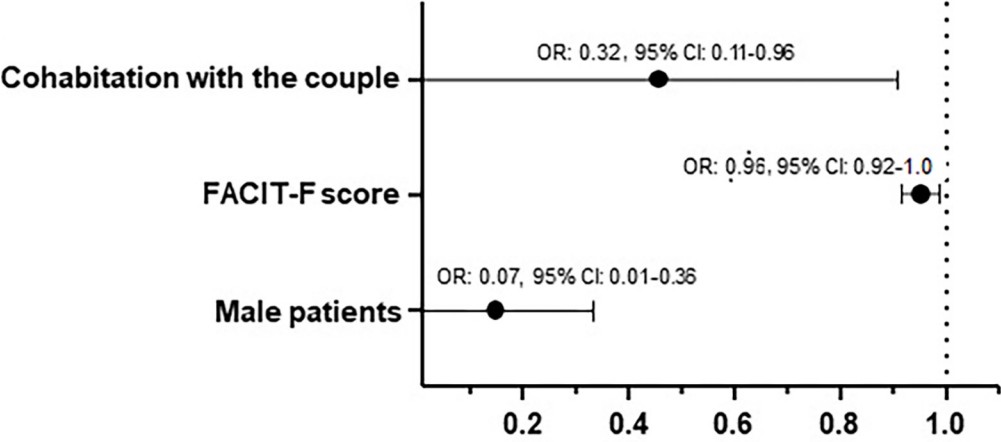

**Fig 3. Results from the multivariable logistic regression analysis: Significant factors associated with ISF.**

may be conditioned/halted during childbearing age, pregnancy, and lactation by the potential teratogenicity and the effects on female fertility [72]. Meanwhile, gender, which refers to the non-physiological components of sex regarded as appropriate to males and females from a sociocultural point of view, is intimately connected to behaviors and actions that impact health and access to healthcare and might be considered a determinant of SH and RH, particularly for women [73]. Our results related to the FACIT-F score agree with previous reports and literature reviews that associated fatigue with ISF in males [6,7,11,19] and females [6,11,12,57]. Fatigue impacts sexual function by decreasing sexual drive and interfering with sexual intercourse [74], while it might also express mental health comorbidity [75]. Finally, the fact that cohabitation with the couple was protective for ISF indicates that it might offer compensatory factors, such as more frequent sexual opportunities, that offset the adverse effects of the rheumatic disease [61]. The partners also have a critical role in influencing well-being, especially for those coping with chronic illness. Most RA patients perceived significant support from their partners [76], and spouses' responses are crucial in promoting adaptation in individuals with chronic pain [77]. These are expressions of the dyadic coping with RA consequences on patients' biography that might positively affect patients' intimacy and sexuality. We could not replicate previous observations in female Mexican patients that suggested an association between menopause and sexual dysfunction, with methotrexate use being protective [60]. Our study included male participants (affecting disease activity), variations in socio-demographics (age, cohabitation, working subpopulation), clinical characteristics (disease duration, disease activity level, and prevalence of STIs), and treatment (prednisone use, number of DMARDs), which may explain the conflicting results. Moreover, the results of regression analysis are influenced by the variables included in the models, which differed between both studies.

It is intriguing that factors related to the COVID-19 environment did not impact patients' sexual function. A recent systematic review and metanalysis of 134 cohorts, aimed at synthesizing mental health outcomes before and during the COVID-19 pandemic, found that at a population level, rather than a mental health crisis, there has been a high level of resilience during COVID-19 and changes in general mental health, anxiety symptoms, and depression symptoms have been minimal to small [78].

The study has some limitations that need to be addressed. Some of these limitations affect the external validity of our findings. Firstly, the participants were recruited from a single academic center in an urban area, which may not represent a diverse population of RA patients. Secondly, the participants primarily had long-standing diseases, and there was an underrepresentation of men. Thirdly, the assessment of sexual function required at least one sexual intercourse in the last four weeks, which may not be applicable to all participants. Additionally, the methods used to assess SF may be biased towards cisgender heterosexual patients. There are also limitations that affect the internal validity of the results. The study was unpowered to detect further differences between male and female participants that might impact SF. Furthermore, the questionnaires used in the study have been validated in Spanish but have not been culturally adapted for the Mexican population. Missing data for some variables reached up to 20%, and no imputation was performed. Lastly, all the questionnaires used were self-reported, which may introduce information, recall, and social desirability biases.

## Conclusions

The study was conducted on a well-defined group of RA patients, reflecting real-life outpatients. All participants underwent comprehensive standardized rheumatologic assessments, including evaluations of their physical and mental health, as well as their perceptions related to COVID-19. These assessments were crucial to achieving the study's objectives and bolstering

both the internal and external validity of the findings. We observed a disproportionate burden of ISF among women with RA during COVID-19 compared to RA male participants. Male sex, lesser fatigue, and cohabitation with the couple were protective against ISF. Regardless of the prevalent use of teratogenic medications, contraceptive use was suboptimal.

Patient-centered care is considered the best healthcare model for patients with rheumatic diseases. Patients with RA have expressed unmet SH and RH needs, and they want their rheumatologists to address these needs in a holistic manner. As specialists, there is an opportunity for research to develop evidence-based SH and RH care strategies that take into consideration the values, preferences, and life circumstances of patients. It's important to incorporate biopsychosocial approaches to data analysis, especially among populations typically underrepresented in the scientific arena.

## Supporting information

**S1 Appendix. STROBE checklist for cross sectional studies.**
(PDF)

**S2 Appendix. COVID-19 survey.**
(PDF)

## Author Contributions

**Conceptualization:** Loraine Ledón-Llanes, Irazú Contreras-Yáñez, Guillermo Arturo Guaracha-Basáñez, Salvador Saúl Valverde-Hernández, Maximiliano Cuevas-Montoya, Ana Belén Ortiz-Haro, Virginia Pascual-Ramos.

**Formal analysis:** Loraine Ledón-Llanes, Irazú Contreras-Yáñez, Virginia Pascual-Ramos.

**Investigation:** Salvador Saúl Valverde-Hernández, Maximiliano Cuevas-Montoya, Ana Belén Ortiz-Haro, Virginia Pascual-Ramos.

**Methodology:** Loraine Ledón-Llanes, Irazú Contreras-Yáñez, Guillermo Arturo Guaracha-Basáñez, Virginia Pascual-Ramos.

**Software:** Irazú Contreras-Yáñez.

**Supervision:** Loraine Ledón-Llanes, Virginia Pascual-Ramos.

**Validation:** Loraine Ledón-Llanes, Irazú Contreras-Yáñez, Guillermo Arturo Guaracha-Basáñez, Virginia Pascual-Ramos.

**Writing – original draft:** Virginia Pascual-Ramos.

**Writing – review & editing:** Loraine Ledón-Llanes, Irazú Contreras-Yáñez, Guillermo Arturo Guaracha-Basáñez, Salvador Saúl Valverde-Hernández, Maximiliano Cuevas-Montoya, Ana Belén Ortiz-Haro.

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
