## [Decision Letter · Decision Letter 0]

12 Feb 2024

PONE-D-23-10059Sexual and reproductive health in Mexican patients with rheumatoid arthritis during the COVID-19 pandemic: A comprehensive approach from the biopsychosocial path.PLOS ONE

Dear Dr. Pascual-Ramos,

Thank you for submitting your manuscript to PLOS ONE. After careful consideration, we feel that it has merit but does not fully meet PLOS ONE’s publication criteria as it currently stands. Therefore, we invite you to submit a revised version of the manuscript that addresses the points raised during the review process.

We look forward to receiving your revised manuscript.

Kind regards,

Milad Khorasani, PhD

Academic Editor

PLOS ONE

Journal Requirements:

Reviewers' comments:

Reviewer's Responses to Questions

**Comments to the Author**

1. Is the manuscript technically sound, and do the data support the conclusions?

Reviewer #1: Yes

Reviewer #2: Partly

2. Has the statistical analysis been performed appropriately and rigorously? 

Reviewer #1: Yes

Reviewer #2: No

3. Have the authors made all data underlying the findings in their manuscript fully available?

Reviewer #1: Yes

Reviewer #2: Yes

4. Is the manuscript presented in an intelligible fashion and written in standard English?

Reviewer #1: No

Reviewer #2: Yes

5. Review Comments to the Author

Reviewer #1: I have had the opportunity to review the manuscript entitled “Sexual and reproductive health in Mexican patients with rheumatoid arthritis during the COVID-19 pandemic: A comprehensive approach from the biopsychosocial path. “ which seems to me to address a relevant, relatively unexplored topic that would be very useful in the comprehensive clinical approach to patients with rheumatoid arthritis, and which seems appropriate to be published in Plos One.

However, although the authors approach the study phenomenon with a pertinent methodological approach, I consider that the manuscript must be improved in a series of aspects of form and substance in order to be accepted.

Below, I expose this series of aspects that I consider should be corrected or refuted with solid scientific arguments by the authors:

First of all, I recommend that the manuscript be reviewed and corrected by a native English speaker, since there are some paragraphs that are difficult to understand due to the writing style used.

Title and Abstract:

A first observation is the absence of correlation or inconsistency between the title and the actual content of the study. The title emphasizes the relevance of linking the central phenomena studied with the concurrence of the SARS-Covid 2 pandemic, however, while reading the manuscript I did not find any solid connection to justify these relationships, except the temporal concurrence of the conducting the study with the pandemic, this leads me to the following question:

Did the authors justify that the presence of the pandemic would have an effect on the sexual and reproductive health of patients with RA? In my opinion, I consider that in order to justify the inclusion of this factor in the title, a comparison of the behavior of the dependent variables of sexual function or health should have been carried out before or after the end of the pandemic.

The title also gives weight to the phenomenon of “reproductive health”, however this variable is barely explored or is explored with unstructured measurement instruments, since the instruments used for the core part of the study (sexual function) are specifically focused. to the domain of function (not health, which is broader from the point of view of semantic content) sexual, not reproductive.

Additionally, from the title and throughout the manuscript the authors use “sexual health” and “sexual function” interchangeably as synonyms, which are not exactly synonymous terms. I consider that for scientific accuracy the authors should standardize the concepts used throughout. of the manuscript.

Material and methods:

A fundamental problem is the risk of selection bias that the study has in relation to the population included, since the patients studied were defined as having RA “based on the criteria of the treating rheumatologist.” Since its verification does not impose much additional effort to carry out the study, it is currently very difficult to justify a research project in RA based on patients who have not met the EULAR/ACR classification criteria. If it is not possible to modify it, this aspect (potential selection bias) should be included and discussed in the limitations of the study.

An important question to evaluate the appearance of measurement biases: Were the various instruments used, in addition to having been translated into Spanish, transculturated for the Mexican population?

In the paragraph dedicated to the operational definition of the sexual function variable in men: Were the cut-off points defined to categorize the presence of impaired sexual function arbitrary definitions made by the authors for the purposes of the study?; Or have they been previously validated?

In this same paragraph,: given that the acronym ISF ¿meaning impaired sexual function? is the first time it is used in the manuscript, it should be expressed non-acronymized, followed by its acronym in parentheses. The same observation applies to the SF acronym in the subtitle of this paragraph.

Another important point that the authors should reflect on is that the instrument used to evaluate sexual function in men has the specific objective of evaluating the phenomenon of erectile dysfunction, which, although similar, is not exactly the same as sexual function. Depending on sexual preference, a person could have decreased erectile function, but be sexually satisfied with this degree of erectile function. This aspect is also susceptible to being included in the limitations of the study.

An important aspect is that in the Methods section the variable “(past or present) sexual discomfort” is not operationally defined, althaugh is used repeatedly in the results section.

Discussion:

It seems too extensive to me, covering aspects of descriptive information of data that are not relevant to the fundamental aspect of the study: the factors associated with altered sexual health in patients with RA, which only occupies 1.5 of the 5.5 pages dedicated to this section of the manuscript.

Since it was carried out on a population from the same country, the authors should include in the introduction and discussion the paper: Alvarez-Nemegyei J, Cervantes-Díaz MT, Avila-Zapata F, Marín-Ordóñez J. Pregnancy outcomes before and after the onset of rheumatoid arthritis. Rev Med Inst Mex Seguro Soc. 2011 Nov-Dec;49(6):599-604, which addresses reproductive health topics in Mexican women with RA.

In relation to the results of the logistic regression analysis carried out, I consider that, instead of being presented as a Forrest-type graph presenting only the variables that reached statistical significance, it should be presented in tabular form with all the variables included in the model, each one with its OR, 95% CI and corresponding p value.

Reviewer #2: This is a very interesting manuscript that includes a subsection of the population that deserves more attention in research. I congratulate them on their efforts.

there are some topics though that I think could improve the manuscript:

In general, The authors indicate that the variables that are affected by RA include non SRH factors. However there are measures that are reported that are more psychosocial factors: The HADS (anxiety and depression), the RAPID-3 (disease activity/severity), the EQ-5D (health related-quality of-Life), the FACIT-F (fatigue) (which I do understand is part of the listed ´biopsychosocial path´, however this can be simplified.

Please use ´participants´ not ´patients´ throughout.

Title: should better reflect what is being studied/found in a more simplified way: one example could be Psychosocial factors associated with sexual function of male and female Rheumatoid Arthritis patients of a. xxxx hospital in Mexico City (or something to that effect)

Abstract:

The writing of the abstract can be improved, especially in the results section to correctly describe results.

What is impaired SF? The defintion is not adequate to understand what you present in results.

The dates seem to be missing when the study was undertaken.

What is ´multiple logistic regression´? Do you mean Multivariable? You do not present multivariable results, nor do you indicate what is being controlled for. You present bivariable results (at least in the abstract…). Were there no significant results with multivariable? If so this is important to state.

In the results section, the results need to be presented more clearly. Example: I do not understand this sentence: ´ Overall, 49.6% of the patients referred to the current family planning method, and 89.7% had children, with a median of 2´.

The measures that are undertaken with males and females (FSFI, satisfaction) should be presented by sex.

It wouldn´t be that ´erectile dysfunction was present in 29.4% of males, but instead 29.4% of males reported erectile dysfunction. You´re are not actually measuring the erectile dysfunction but instead the report.

´Male` is not a gender, it is sex.

The OR and p values can be truncated at two numbers after the decimal.

% results should be presented with OR values.

Could there be confounding variables that need to be controlled for (age)?

Conclusions:

You cannot conclude that RA compromises SRH, you have no control group (non RA). All you can say that some psychosocial factors were found in individuals with RA, and there were differences of participant sex

Introduction:

You state that WHO recommends using SH (which includes RH), yet you continue to use SRH. Why is this?

Authors write: ´RH in RA patients has focused on reduced fertility (30-35) which has been related to the disease process itself [30], therapy [29], gonadal dysfunction [36], physicians’ advice [30, 31], and individual decisions [30, 31]. ´ what do you mean by physicians´advice and individual decisions?

This last sentence and the following starting line 86 need to be put together into one. You´ve already said reduced fertility (subfertility should be called ´infertility´ . This one item (infertility) can be included in the previous sentence as ´infertility´ and put the citations together.

*Methods*

The study was undertaken between 2020 and 2022. Were there still covid instruments being included in 2022?

No need to cite STROBE and include it in as supplementary material. I believe it is just the journal that needs this when submitting.

The methods include qualitative research. These results are not reflected in the abstract.

How were the quantitative questionnaires administered? Paper format? Electronic?

In the definitions and groups (what is ´groups´?) you only mention impaired sexual function for males and females. This is not a ´reproductive health´ measure per se, but instead by definition, a sexual health measure. Perhaps think about this (as these are you outcome variables) when reworking your title and focus in your abstract.

Sample size: if you only needed 151 patients, why did you include more than 100 more?

what is the overall population estimated with RA in CDMX?

Results seem to be expressed (at least in the abstract as OR not AOR or aOR which would be adjusted).

Furthermore, you do not indicate what you are adjusting for (other than RAPID-3), but surely age must be a confounder when dealing with sexual function (although it is shocking to me- the literature says that starting late 20´s early 30s, women´s sexual function may start to decline!)

Results:

NO need to duplicate in written results what is in the table. This section could be rearranged, considering the recommendations below of the tables.

What is ´middle aged female? This is a vague definition that can mean a lot of things to different people. Median and IQR should be used for age.

The table 1 needs some better structure. For example, many variables are presented with mean or medians? This needs to be defined better. But there are variables like Religious beliefs (what does this mean? Is it ANY religious beliefs? Is it being spiritual? Being in a relationship is the same. So it should be set up as categories for some of these variables: Employment (formal is one category and informal is another category), etc.

Table 2 is perhaps important, but these variables need to described in light of your outcome variables (and included in the abstract and better woven into the paper). Otherwise it´s just information that is hanging in the air. Are the anxiety and depression scales ONLY covid related? Why are they not part of other tables? Where these

Now I see RH results (these seem to be missing sadly from a lot of the paper)

For the results section- I would organize like this:

Table 1: demographic characteristics with medians(IQR), means(SD) and categories correctly presented

Table two: Outcome variables presented of impaired SF and non-impaired SF(columns) against demographic variables (rows). The covid variables should just be other non-outcome variables but labeled correctly.

Table three: OR with significant variables found in table two.

Discussion:

This should be in sync with the more developed results. As it is, it is quite long and wordy, but does not need to be. You need to decide if COVID is going to be main non-outcome variables or not (depending on results), this is not even mentioned in the first paragraph of the discussion.

6. PLOS authors have the option to publish the peer review history of their article (what does this mean?). If published, this will include your full peer review and any attached files.

Reviewer #1: No

Reviewer #2: No

---

## [Author Response · Author response to Decision Letter 0]

26 Feb 2024

Responses to reviewers

Reviewer #1

I have had the opportunity to review the manuscript entitled “Sexual and reproductive health in Mexican patients with rheumatoid arthritis during the COVID-19 pandemic: A comprehensive approach from the biopsychosocial path “which seems to me to address a relevant, relatively unexplored topic that would be very useful in the comprehensive clinical approach to patients with rheumatoid arthritis, and which seems appropriate to be published in Plos One.

However, although the authors approach the study phenomenon with a pertinent methodological approach, I consider that the manuscript must be improved in a series of aspects of form and substance to be accepted.

Response. We appreciate the reviewer's comments.

Below, I expose this series of aspects that I consider should be corrected or refuted with solid scientific arguments by the authors:

First of all, I recommend that the manuscript be reviewed and corrected by a native English speaker, since some paragraphs are difficult to understand due to the writing style used.

Response. The manuscript was already reviewed by an editing service (Grammarly business edition). It has been re-reviewed.

Title and Abstract:

A first observation is the absence of correlation or inconsistency between the title and the actual content of the study. The title emphasizes the relevance of linking the central phenomena studied with the concurrence of the SARS-Covid 2 pandemic, however, while reading the manuscript I did not find any solid connection to justify these relationships, except the temporal concurrence of the conducting the study with the pandemic, this leads me to the following question:

Did the authors justify that the presence of the pandemic would have an effect on the sexual and reproductive health of patients with RA? In my opinion, I consider that in order to justify the inclusion of this factor in the title, a comparison of the behavior of the dependent variables of sexual function or health should have been carried out before or after the end of the pandemic. 

The title also gives weight to the phenomenon of “reproductive health,” however, this variable is barely explored or is explored with unstructured measurement instruments since the instruments used for the core part of the study (sexual function) are specifically focused. to the domain of function (not health, which is broader from the point of view of semantic content) sexual, not reproductive.

Response. We have updated the title of the manuscript to highlight the main results, which are the factors associated with impaired sexual function, and omitted the reference to the COVID-19 pandemic. The second reviewer has a similar request. The title has been updated to “Biopsychosocial factors are associated with impaired sexual function in Mexican patients with rheumatoid arthritis.”

Additionally, from the title and throughout the manuscript the authors use “sexual health” and “sexual function” interchangeably as synonyms, which are not exactly synonymous terms. I consider that for scientific accuracy the authors should standardize the concepts used throughout of the manuscript.

Response. We agree with the reviewer that sexual health and sexual function are not synonyms; in fact, sexual function contributes to sexual health. We have been more accurate about the terms used throughout the manuscript. In particular, in the material and methods and results sections, the term "sexual function" is preferred over the term sexual health, consistent with the outcome of interest, that is, impaired sexual function, defined with appropriate tools. However, the reviewer might also have found the term sexual health throughout the text when appropriate and when mentioned by other authors.

Material and methods:

A fundamental problem is the risk of selection bias that the study has in relation to the population included, since the patients studied were defined as having RA “based on the criteria of the treating rheumatologist.” Since its verification does not impose much additional effort to carry out the study, it is currently very difficult to justify a research project in RA based on patients who have not met the EULAR/ACR classification criteria. If it is not possible to modify it, this aspect (potential selection bias) should be included and discussed in the limitations of the study.

Response. We disagree with the reviewer. 2010 EULAR/ACR criteria are classification criteria and should not be used for RA diagnosis (Aletaha et al, 2010). As the authors state, “While classification criteria are potentially adopted for use as aids for diagnosis, the focus of this endeavor was not on developing diagnostic criteria or providing a referral tool for primary care physicians…The criteria do not remove the onus on individual physicians, especially in the face of unusual presentations, to reach a diagnostic opinion that might vary from the assignment obtained using the criteria…. Much like other classification criteria, clinicians may be able to diagnose an individual who has not met the classification criteria definition or who has features that are not a component of the classification criteria.” Aletaha et al. also stated in their manuscript that “there is no gold standard for RA diagnosis.” Most of the time, in daily practice, rheumatologists establish the diagnosis based on the combination of clinical signs/symptoms, available clinical tests, and knowledge about the epidemiology of the area. (Aggarwal R et al. Distinctions Between Diagnostic and Classification Criteria? Arthritis Care Res (Hoboken). 2015 July ; 67(7): 891–897. doi:10.1002/acr.22583). Finally, it is generally accepted that the diagnosis of rheumatic diseases should be individualized because the classification criteria could not include all the phenotypes of diseases, including unusual features or presentations. Diagnosed by board-certified rheumatologists has been used (Young KA et al. Arthritis Rheum 2013). 

An important question to evaluate the appearance of measurement biases: Were the various instruments used, in addition to having been translated into Spanish, transculturated for the Mexican population?

Response. FSFI and IIEF have not undergone transcultural adaptation for the Mexican population. We have addressed it as a limitation. 

In the paragraph dedicated to the operational definition of the sexual function variable in men: Were the cut-off points defined to categorize the presence of impaired sexual function arbitrary definitions made by the authors for the purposes of the study?; Or have they been previously validated?

Response. We used previously published cut-offs. 

In this same paragraph, given that the acronym ISF ¿meaning impaired sexual function? This is the first time it is used in the manuscript. It should be expressed non-acronymized, followed by its acronym in parentheses. The same observation applies to the SF acronym in the subtitle of this paragraph.

Response. We are sorry for the mistake. We have reviewed the updated version to address the point raised by the reviewer. 

Another important point that the authors should reflect on is that the instrument used to evaluate sexual function in men has the specific objective of evaluating the phenomenon of erectile dysfunction, which, although similar, is not exactly the same as sexual function. Depending on sexual preference, a person could have decreased erectile function, but be sexually satisfied with this degree of erectile function. This aspect is also susceptible to being included in the study's limitations.

Response. We agree with the reviewer and have addressed the point as a limitation. 

An important aspect is that in the Methods section, the variable “(past or present) sexual discomfort” is not operationally defined, although it is used repeatedly in the Results section.

Response. We agree with the reviewer, and in the updated version, we provide an operational definition of sexual discomforts and a reference. 

Discussion:

It seems too extensive to me, covering aspects of descriptive information of data that are not relevant to the fundamental aspect of the study: the factors associated with altered sexual health in patients with RA, which only occupies 1.5 of the 5.5 pages dedicated to this section of the manuscript.

Response. We have updated the discussion section. 

Since it was carried out on a population from the same country, the authors should include in the introduction and discussion the paper: Alvarez-Nemegyei J, Cervantes-Díaz MT, Avila-Zapata F, Marín-Ordóñez J. Pregnancy outcomes before and after the onset of rheumatoid arthritis. Rev Med Inst Mex Seguro Soc. 2011 Nov-Dec;49(6):599-604, which addresses reproductive health topics in Mexican women with RA.

Response. The suggestion has been adopted. 

In relation to the results of the logistic regression analysis carried out, I consider that, instead of being presented as a Forrest-type graph presenting only the variables that reached statistical significance, it should be presented in tabular form with all the variables included in the model, each one with its OR, 95% CI and corresponding p value.

Response. The suggestion has been adopted. In addition to the Forrest-type graph (we have conserved as it summarizes the test-based backward selection that was used to define the final model), we have added Table 3, which presents OR and aOR (required by one reviewer) for all the variables included in the model. 

Reviewer #2

This is a very interesting manuscript that includes a subsection of the population that deserves more attention in research. I congratulate them on their efforts.

there are some topics, though, that I think could improve the manuscript:

Response. We appreciate the reviewer's comments.

In general, The authors indicate that the variables that are affected by RA include non SRH factors. However, there are measures that are reported that are more psychosocial factors: The HADS (anxiety and depression), the RAPID-3 (disease activity/severity), the EQ-5D (health related-quality of-Life), the FACIT-F (fatigue) (which I do understand is part of the listed ´biopsychosocial path´, however this can be simplified.

Response. We have tried to provide an updated version that is more accurate and highlights relevant results. 

Please use ´participants,´ not ´patients´ throughout.

Response. We have adopted the suggestion, particularly in the methods and results sections. 

Title: should better reflect what is being studied/found in a more simplified way: one example could be Psychosocial factors associated with sexual function of male and female Rheumatoid Arthritis patients of a. xxxx hospital in Mexico City (or something to that effect)

Response. We have followed the reviewer's suggestion and updated the title to "Biopsychosocial factors are associated with impaired sexual function in Mexican patients with rheumatoid arthritis."

Abstract:

The writing of the abstract can be improved, especially in the results section to correctly describe results.

Response. We have updated the abstract to address the reviewer´s point. 

What is impaired SF? The definition is not adequate to understand what you present in the results.

Response. We have provided a better definition of ISF. 

The dates seem to be missing when the study was undertaken.

Response. We have updated the dates. 

What is ´multiple logistic regression´? Do you mean Multivariable? You do not present multivariable results, nor do you indicate what is being controlled for. You present bivariable results (at least in the abstract…). Were there no significant results with multivariable? If so this is important to state.

Response. We have updated the term multiple to the appropriate term, multivariable. We have updated the section to make the results from the multivariable regression analysis more evident. 

In the results section, the results need to be presented more clearly. Example: I do not understand this sentence: Overall, 49.6% of the patients referred to the current family planning method, and 89.7% had children, with a median of 2´.

Response. We have updated the whole section. 

The measures that are undertaken with males and females (FSFI, satisfaction) should be presented by sex.

Response. The suggestion has been adopted. 

It wouldn´t be that ´erectile dysfunction was present in 29.4% of males, but instead 29.4% of males reported erectile dysfunction. You´re are not actually measuring the erectile dysfunction but instead the report.

Response. The reviewer is right, we have updated the abstract. 

´Male` is not a gender, it is sex.

Response. We have updated the term. 

The OR and p values can be truncated at two numbers after the decimal.

Response. The suggestion has been adopted. 

% results should be presented with OR values.

Response. We are unsure about what the reviewer is asking for, however, we have added OR for the explanatory variables of interest in table 3.

Could there be confounding variables that need to be controlled for (age)?

Response. We added age as a confounder variable in Table 3. 

Conclusions:

You cannot conclude that RA compromises SRH, you have no control group (non RA). All you can say that some psychosocial factors were found in individuals with RA, and there were differences of participant sex

Response. The conclusions have been updated. 

Introduction:

You state that WHO recommends using SH (which includes RH), yet you continue to use SRH. Why is this?

Response. We agree with the reviewer. We have omitted the acronym SRH and used/assessed as recommended, SH and RH throughout the text.

Authors write: ´RH in RA patients has focused on reduced fertility (30-35) which has been related to the disease process itself [30], therapy [29], gonadal dysfunction [36], physicians’ advice [30, 31], and individual decisions [30, 31]. ´ what do you mean by physicians advice and individual decisions?

Response. We have updated the sentence to improve clarity.

This last sentence and the following starting line 86 need to be put together into one. You´ve already said reduced fertility (subfertility should be called ´infertility´ . This one item (infertility) can be included in the previous sentence as ´infertility´ and put the citations together.

Response. We have updated the sentence according to the reviewer´s suggestion. 

*Methods*

The study was undertaken between 2020 and 2022. Were there still covid instruments being included in 2022?

Response. The COVID-19 survey was administered to all the participants. 

There is no need to cite STROBE and include it as supplementary material. I believe it is just the journal that needs this when submitting.

Response. We are including STROBE as the journal recommends it should be used in observational studies. 

The methods include qualitative research. These results are not reflected in the abstract.

How were the quantitative questionnaires administered? Paper format? Electronic?

Response. Data related to RH in the abstract was derived from the semi-structured interview. Questionnaires were administered in paper format. It has been updated in the corresponding section. 

In the definitions and groups (what is ´groups´?) you only mention impaired sexual function for males and females. This is not a ´reproductive health´ measure per se, but instead by definition, a sexual health measure. Perhaps think about this (as these are you outcome variables) when reworking your title and focus in your abstract.

Response. We have renamed the paragraph as “Definitions” and added one more definition (“sexual discomforts”) as requested by the second reviewer. We have updated the abstract, the manuscript, and the title to highlight factors associated with the outcome of interest, ISF in males and females. 

Sample size: if you only needed 151 patients, why did you include more than 100 more?

what is the overall population estimated with RA in CDMX?

Response. The sample size estimation was based on a hypothesis considering the RAPID-3 effect size odds ratio of 2.5 in a non-normal distribution, using a one-tailed test, 5% significance level, and 80% power. However, we assumed other factors could be associated with ISF, and we might need a bigger sample size to guarantee suffi

---

## [Decision Letter · Decision Letter 1]

1 Apr 2024

PONE-D-23-10059R1Biopsychosocial factors are associated with impaired sexual function in Mexican patients with rheumatoid arthritis.PLOS ONE

Dear Dr. Pascual-Ramos,

Thank you for submitting your manuscript to PLOS ONE. After careful consideration, we feel that it has merit but does not fully meet PLOS ONE’s publication criteria as it currently stands. Therefore, we invite you to submit a revised version of the manuscript that addresses the points raised during the review process.

We look forward to receiving your revised manuscript.

Kind regards,

Milad Khorasani, PhD

Academic Editor

PLOS ONE

Reviewers' comments:

Reviewer's Responses to Questions

**Comments to the Author**

1. If the authors have adequately addressed your comments raised in a previous round of review and you feel that this manuscript is now acceptable for publication, you may indicate that here to bypass the “Comments to the Author” section, enter your conflict of interest statement in the “Confidential to Editor” section, and submit your "Accept" recommendation.

Reviewer #1: All comments have been addressed

Reviewer #2: All comments have been addressed

2. Is the manuscript technically sound, and do the data support the conclusions?

Reviewer #1: Yes

Reviewer #2: Partly

3. Has the statistical analysis been performed appropriately and rigorously? 

Reviewer #1: Yes

Reviewer #2: Yes

4. Have the authors made all data underlying the findings in their manuscript fully available?

Reviewer #1: Yes

Reviewer #2: Yes

5. Is the manuscript presented in an intelligible fashion and written in standard English?

Reviewer #1: Yes

Reviewer #2: Yes

6. Review Comments to the Author

Reviewer #1: I consider that the authors have carried out all the suggested corrections, or have refuted in a solid argumentative manner those with which they did not agree, so I consider that the manuscript could be accepted.

However, before being accepted there are some specific aspects that require correction or clarification:

The aOR values of the predictor variables that were significant in the logistic regression model that are mentioned in the abstract, in the text of the results and in Figure 3 do not coincide with those shown in Table 3. In fact, none of the aOR and p values in the corresponding column are significant.

There appears to be an error in placing the value of the decimal point in the confidence interval of the FACIT variable in Figure 3, it is mentioned as (0.92-0.10), when in reality its correct value appears to be (0.92-1.00).

I consider that once these specific situations have been corrected or clarified, the article can be accepted.

Reviewer #2: Thank you for the opportunity to review this manuscript. Although it is much improved from the last version, I still feel there are things that can be improved upon.

Abstract

- In the intro (line 83) you say there are characteristics for the LA region (greater female: male ratio and younger age at presentation, although you show this in the intro (with a citation), this sentence alone in the abstract is a bit distracting. Perhaps an opening statement should be that RA affects sexual function or something more general as you are not focused on comparing LA patients to others in the world but instead the affects on

- The abstract still indicates ´patients´ (the term participants was recommended to use). No need to mention more than the term ´Methods´ in the subsection.

- Idncate in the methods section if questionnaires were provider-administered? Study-team administered? self-administered?

- No need to say 1st and 31st (you can just say September 1 yyyy -January 31 yyyy (same with Methods section in body of document)

- Descriptive statistics and multivariable logistic regression analysis were used. (were used to…? (to study x factors related to y)

- What is FACIT-F.? Please don´t use acronyms without defining.

- The conclusion needs to be more explicit. X, y and z were found to be associated with ISF…. We recommend….or a conclusion of the findings.

Introduction:

- You cite Caucasians and which sex is most affected (please correct to females and males as gender is not as much a factor as sex is)- can you find a citation that looks at the F:M ratio among Latin Americans?

Methods

- Normally the ethics section goes at the end of the Methods section

- It is not clear (line 136) who applied the paper survey. They say administered, how was that administration (this should be discussed in the limitations section as well as any way you apply a survey there are bias’s to how people respond)

- Definitions subtitle could be better titled as ´Measures´

- ´Statistical analysis´ subtitle could be better called ´Statistical Analyses´ as there are more than one.

- I think the terms univariable/multivariable or univariate/multivariate should be homogenized. Right now, there is univariate/multivariable.

- What was the adjustment in the aOR for? (i.e. what did you adjust for)

- Missing data : (line 222) is this only per question? Or if someone said I don´t want to answer to one question then none of their questions were included?

- Line 226´A value of p<0.05 was considered statistically significant.´ seems wrongly placed

Results

- Line 230 missing data needs to be better explained as above (and no need to repeat if this has already been explained in methods)

- Table 1: what are ´self-referring Catholics and self-referring non-religious?´ is this self-reported? If so, could probably drop it as would think all religion is self-reported?

- This table could be better presented as

Variable N=xxx

Category n % or median(IQR)

Category n

category n

And some variables could be kept as one (e.g. age)

Please indicate what the * stands for

Table 1 should be presented with male and females as well:

All participants Male participants Female participants

Variable N=xxx % or median(IQR) (should be denoted as median if you present median) N=xxx % or median(IQR) (should be denoted as median if you present median) N=xxx % or median(IQR) (should be denoted as median if you present median)

Category n

Category n

category n

Please don´t repeat written results if they are in the table, unless they are some of the main results. The written results can be cut down a lot I think.

Table three could be presented as two extra columns of table 2 (in the same format as my suggestion for Table 1 above), as its best to be able to see OR and aOR against %. Please define what the current table 2 p-values are describing.

Figures 1 and 2: make sure data is not repeated in tables. If it is, please eliminate figures.

Figures 3 (it´s not titled but I think it is figure 3) doesn´tadd much to the data as this is already presented in the table.

Was age added in the model as an adjusting factor? This should be explained in the methods perhaps.

Discussion.

The first paragraph should highlight a few findings from your analyses and the following paragraphs explain these findings further and compare to other research in the same paragraph.

Please check grammar throughout but (line 340, 341 especially)

Your discussion on male participants (lines aprox. 350) is biased because of your low sample size, this should be highlighted.

Fully discuss limitations (some ideas above)

7. PLOS authors have the option to publish the peer review history of their article (what does this mean?). If published, this will include your full peer review and any attached files.

Reviewer #1: **Yes: **José Alvarez-Nemegyei

Reviewer #2: No

---

## [Author Response · Author response to Decision Letter 1]

8 Apr 2024

Reviewer #1

I consider that the authors have carried out all the suggested corrections, or have refuted in a solid argumentative manner those with which they disagreed, so I consider that the manuscript could be accepted.

Response. We appreciate the reviewer's comments.

However, before being accepted, there are some specific aspects that require correction or clarification:

The aOR values of the predictor variables that were significant in the logistic regression model that are mentioned in the abstract, in the text of the results and in Figure 3 do not coincide with those shown in Table 3. In fact, none of the aOR and p values in the corresponding column are significant.

Response. In the previous revision, the reviewer suggested a table (currently table 3) with OR and aOR (95% CI and p values) from all the variables included in the logistic regression model, and not only those that reached statistical significance. We have updated the title of Table 3 to better align with the information provided. In addition, aORs presented in the results section of the abstract are aligned with data from Figure 3. Both summarize only the variables from the multivariable regression model that ended up reaching statistical significance. 

There appears to be an error in placing the value of the decimal point in the confidence interval of the FACIT variable in Figure 3, it is mentioned as (0.92-0.10), when in reality its correct value appears to be (0.92-1.00).

I consider that once these specific situations have been corrected or clarified, the article can be accepted.

Response. We are sorry for the mistake. We have updated the Figure 3.

Reviewer #2: 

Thank you for the opportunity to review this manuscript. Although it is much improved from the last version, I still feel there are things that can be improved upon.

Abstract

- In the intro (line 83) you say there are characteristics for the LA region (greater female: male ratio and younger age at presentation, although you show this in the intro (with a citation), this sentence alone in the abstract is a bit distracting. Perhaps an opening statement should be that RA affects sexual function or something more general as you are not focused on comparing LA patients to others in the world but instead the affects on.

Response. We have adopted the suggestion and proposed an updated version. 

- The abstract still indicates ´patients´ (the term participants was recommended to use). No need to mention more than the term ´Methods´ in the subsection.

Response. We have adopted both suggestions. 

- Indicate in the methods section if questionnaires were provider-administered? Study-team administered? self-administered?

Response. We have updated the abstract and clarified that questionnaires were self-administered. 

- No need to say 1st and 31st (you can just say September 1 yyyy -January 31 yyyy (same with Methods section in the body of document)

Response. We have adopted the suggestion. 

- Descriptive statistics and multivariable logistic regression analysis were used. (were used to…? (to study x factors related to y)

Response. We have updated the sentence to adopt the suggestion. 

- What is FACIT-F.? Please don´t use acronyms without defining it.

Response. We apologize. We have defined FACIT-F. 

- The conclusion needs to be more explicit. X, y and z were found to be associated with ISF…. We recommend….or a conclusion of the findings.

Response. We have updated the conclusions following the reviewer´s recommendation. 

Introduction:

- You cite Caucasians and which sex is most affected (please correct to females and males as gender is not as much a factor as sex is)- can you find a citation that looks at the F:M ratio among Latin Americans?

Response. We have adopted the suggestion. Ref 38 is appropriately cited. Although the paper focuses on treatment, it describes the data from 1093 patients from 14 LATAM countries, of whom 85.3% were females. 

Methods

- Normally the ethics section goes at the end of the Methods section

Response. The ethics section was moved to the end of the methods section.

- It is not clear (line 136) who applied the paper survey. They say administered, how was that administration (this should be discussed in the limitations section as well as any way you apply a survey there are bias’s to how people respond)

Response. All questionnaires were self-administered, and this information was updated as suggested. We have added biases associated with self-reported questionnaires to the limitations section. 

- Definitions subtitle could be better titled as ´Measures´

Response. We have updated the subtitle.

- ´Statistical analysis´ subtitle could be better called ´Statistical Analyses´ as there are more than one.

Response. We have updated the subtitle.

- I think the terms univariable/multivariable or univariate/multivariate should be homogenized. Right now, there is univariate/multivariable.

Response. We have adopted the suggestion. 

- What was the adjustment in the aOR for? (i.e. what did you adjust for)

Response. We have updated the information required in the statistical analyses section. 

- Missing data : (line 222) is this only per question? Or if someone said I don´t want to answer to one question then none of their questions were included?

Response. We have updated the paragraph.

- Line 226´A value of p<0.05 was considered statistically significant´ seems wrongly placed

Response. We have updated the paragraph. 

Results

- Line 230 missing data needs to be better explained as above (and no need to repeat if this has already been explained in methods)

Response. We have omitted the information from the results section and provided a better explanation in the methods section. 

- Table 1: what are ´self-referring Catholics and self-referring non-religious?´ is this self-reported? If so, could probably drop it as would think all religion is self-reported?

Response. We agree with the reviewer and have omitted the information.

- This table could be better presented as

Variable N=xxx

Category n % or median(IQR)

Category n

category n

And some variables could be kept as one (e.g. age)

Please indicate what the * stands for

Table 1 should be presented with male and females as well:

All participants Male participants Female participants

Variable N=xxx % or median(IQR) (should be denoted as median if you present median) N=xxx % or median(IQR) (should be denoted as median if you present median) N=xxx % or median(IQR) (should be denoted as median if you present median)

Category n

Category n

category n

Response. We have modified Table 1 and incorporated all the suggestions. 

Please don´t repeat written results if they are in the table, unless they are some of the main results. The written results can be cut down a lot I think.

Response. We have reviewed the whole section, and currently, there are no results duplicated in the text and the tables or figures. 

Table three could be presented as two extra columns of table 2 (in the same format as my suggestion for Table 1 above), as its best to be able to see OR and aOR against %. 

Response. Table 3 responds to reviewer one request to present OR and aOR for ALL the variables included in the multivariable logistic regression model (instead of presenting only those that ended up being statistically significant, which are summarized in Figure 3). For this reason, Table 3 should remain. 

Please define what the current table 2 p-values are describing.

Response. Table 2 has been updated, and the reviewer's request has been included in the text. 

Figures 1 and 2: make sure data is not repeated in tables. If it is, please eliminate figures.

Response. In the current version, there are no repeated data. 

Figure 3 (it´s not titled, but I think it is figure 3) doesn t add much to the data as this is already presented in the table.

Response. We consider Figure 3 should remain as it presents the results (variables that ended up being statistically significant) from the multivariable regression analysis. These are not presented elsewhere. 

Was age added in the model as an adjusting factor? This should be explained in the methods perhaps.

Response. Yes, it was, and we have clarified it. 

Discussion.

The first paragraph should highlight a few findings from your analyses and the following paragraphs explain these findings further and compare to other research in the same paragraph.

Response. We have organized the discussion as suggested. 

Please check grammar throughout but (line 340, 341 especially).

Response. We have used an editing service (Grammarly Business). 

Your discussion on male participants (lines aprox. 350) is biased because of your low sample size, this should be highlighted.

Response. We agree with the reviewer and have updated the paragraph. We have also assumed that the number of male participants is a limitation of the study. 

Fully discuss limitations (some ideas above)

Response. We have updated the limitations section.

---

## [Decision Letter · Decision Letter 2]

28 May 2024

PONE-D-23-10059R2Biopsychosocial factors are associated with impaired sexual function in Mexican patients with rheumatoid arthritis.PLOS ONE

Dear Dr. Pascual-Ramos,

Thank you for submitting your manuscript to PLOS ONE. After careful consideration, we feel that it has merit but does not fully meet PLOS ONE’s publication criteria as it currently stands. Therefore, we invite you to submit a revised version of the manuscript that addresses the points raised during the review process.

We look forward to receiving your revised manuscript.

Kind regards,

Milad Khorasani, PhD

Academic Editor

PLOS ONE

Journal Requirements:

Reviewers' comments:

Reviewer's Responses to Questions

**Comments to the Author**

1. If the authors have adequately addressed your comments raised in a previous round of review and you feel that this manuscript is now acceptable for publication, you may indicate that here to bypass the “Comments to the Author” section, enter your conflict of interest statement in the “Confidential to Editor” section, and submit your "Accept" recommendation.

Reviewer #1: All comments have been addressed

Reviewer #3: (No Response)

2. Is the manuscript technically sound, and do the data support the conclusions?

Reviewer #1: Yes

Reviewer #3: Partly

3. Has the statistical analysis been performed appropriately and rigorously? 

Reviewer #1: Yes

Reviewer #3: Yes

4. Have the authors made all data underlying the findings in their manuscript fully available?

Reviewer #1: Yes

Reviewer #3: Yes

5. Is the manuscript presented in an intelligible fashion and written in standard English?

Reviewer #1: Yes

Reviewer #3: Yes

6. Review Comments to the Author

Reviewer #1: (No Response)

Reviewer #3: Could authors explain the differences regarding a previous paper that explored the prevalence of sexual dysfunction in Mexican women with rheumatoid arthritis (Rojo-Contreras et al. Healthcare (Basel) 2022 Sep 21;10(10):1825. doi: 10.3390/healthcare10101825) focusing on the associated factors of sexual dysfunction to establish the main differences regarding this. In addition to see for the main contributions of the study carried out, author´s should incorporate into the Discussion section the advantages, disadvantages and areas of opportunity for the internal and external validity of the study carried out.

7. PLOS authors have the option to publish the peer review history of their article (what does this mean?). If published, this will include your full peer review and any attached files.

Reviewer #1: **Yes: **Jose Alvarez-Nemegyei

Reviewer #3: No

---

## [Author Response · Author response to Decision Letter 2]

30 May 2024

Responses to reviewers

Could authors explain the differences regarding a previous paper that explored the prevalence of sexual dysfunction in Mexican women with rheumatoid arthritis (Rojo-Contreras et al. Healthcare (Basel) 2022 Sep 21;10(10):1825. doi: 10.3390/healthcare10101825) focusing on the associated factors of sexual dysfunction to establish the main differences regarding this. 

Response. We have included the reference and updated the discussion section to highlight the similarities and differences between both studies.

In addition to see for the main contributions of the study carried out, author´s should incorporate into the Discussion section the advantages, disadvantages and areas of opportunity for the internal and external validity of the study carried out.

Response. In the limitations section, we discussed the internal and external validity of the results. In the conclusion, we summarized the study's advantages and areas of opportunity for future research.

---

## [Decision Letter · Decision Letter 3]

6 Jun 2024

Biopsychosocial factors are associated with impaired sexual function in Mexican patients with rheumatoid arthritis.

PONE-D-23-10059R3

Dear Dr.  Pascual-Ramos,

We’re pleased to inform you that your manuscript has been judged scientifically suitable for publication and will be formally accepted for publication once it meets all outstanding technical requirements.

Kind regards,

Milad Khorasani, PhD

Academic Editor

PLOS ONE

Additional Editor Comments (optional):

Reviewers' comments:

Reviewer's Responses to Questions

**Comments to the Author**

1. If the authors have adequately addressed your comments raised in a previous round of review and you feel that this manuscript is now acceptable for publication, you may indicate that here to bypass the “Comments to the Author” section, enter your conflict of interest statement in the “Confidential to Editor” section, and submit your "Accept" recommendation.

Reviewer #3: All comments have been addressed

2. Is the manuscript technically sound, and do the data support the conclusions?

Reviewer #3: Yes

3. Has the statistical analysis been performed appropriately and rigorously? 

Reviewer #3: Yes

4. Have the authors made all data underlying the findings in their manuscript fully available?

Reviewer #3: Yes

5. Is the manuscript presented in an intelligible fashion and written in standard English?

Reviewer #3: Yes

6. Review Comments to the Author

Reviewer #3: Authors have attended the comments and suggestion, and accordingly them the manuscript in the Discussion section was improved.

7. PLOS authors have the option to publish the peer review history of their article (what does this mean?). If published, this will include your full peer review and any attached files.

Reviewer #3: No

---

## [Editor Report · Acceptance letter]

16 Aug 2024

PONE-D-23-10059R3 

PLOS ONE

Dear Dr. Pascual-Ramos, 

I'm pleased to inform you that your manuscript has been deemed suitable for publication in PLOS ONE. Congratulations! Your manuscript is now being handed over to our production team.

Kind regards, 

on behalf of

Dr. Milad Khorasani 

Academic Editor

PLOS ONE